# Bidirectional Variational Inference for Non-Autoregressive Text-to-Speech

**Yoonhyung Lee, Joongbo Shin, Kyomin Jung**
Department of Electrical and Computer Engineering
Seoul National University
Seoul, South Korea
`{cpi1234, jbshin, kjung}`@snu.ac.kr

## Abstract

Although early text-to-speech (TTS) models such as Tacotron 2 have succeeded in generating human-like speech, their autoregressive architectures have several limitations: (1) They require a lot of time to generate a mel-spectrogram consisting of hundreds of steps. (2) The autoregressive speech generation lacks robustness due to its error propagation property. In this paper, we propose a novel non-autoregressive TTS model called BVAE-TTS, which eliminates the architectural limitations and generates a mel-spectrogram in parallel. BVAE-TTS adopts a bidirectional-inference variational autoencoder (BVAE) that learns hierarchical latent representations using both bottom-up and top-down paths to increase its expressiveness. To apply BVAE to TTS, we design our model to utilize text information via an attention mechanism. By using attention maps that BVAE-TTS generates, we train a duration predictor so that the model uses the predicted duration of each phoneme at inference. In experiments conducted on LJSpeech dataset, we show that our model generates a mel-spectrogram 27 times faster than Tacotron 2 with similar speech quality. Furthermore, our BVAE-TTS outperforms Glow-TTS, which is one of the state-of-the-art non-autoregressive TTS models, in terms of both speech quality and inference speed while having 58% fewer parameters.

## 1 Introduction

End-to-end text-to-speech (TTS) systems have recently attracted much attention, as neural TTS models began to generate high-quality speech that is very similar to the human voice (Sotelo et al., 2017; Wang et al., 2017; Shen et al., 2018; Ping et al., 2018; Li et al., 2019). Typically, those TTS systems first generate a mel-spectrogram from a text using a sequence-to-sequence (seq2seq) model (Sutskever et al., 2014) and then synthesize speech from the mel-spectrogram using a neural vocoder like WaveGlow (Prenger et al., 2019).

Early neural TTS systems have used an autoregressive (AR) architecture to generate a mel-spectrogram mainly because of its two benefits. First, the AR generation eases the difficulty of modeling mel-spectrogram distribution by factorizing the distribution into the product of homogeneous conditional factors in sequential order. Second, the seq2seq based AR architecture helps the model predict the length of the target mel-spectrogram from an input text, which is a non-trivial task because there are no pre-defined rules between the lengths of text and mel-spectrogram.

Although they facilitate high-quality speech synthesis, AR TTS models have several shortcomings. First, they cannot generate a mel-spectrogram in parallel, so the inference time increases linearly with mel-spectrogram time steps. Second, the AR-based generation suffers from accumulated prediction error, resulting in being vulnerable to the out-of-domain data, e.g. very long input text, or text patterns not existing in the training dataset.

In this work, we present a novel non-AR TTS model called BVAE-TTS that achieves fast and robust high-quality speech synthesis. BVAE-TTS generates a mel-spectrogram in parallel by adopting a bidirectional-inference variational autoencoder (BVAE) (Sønderby et al., 2016; Kingma et al., 2016; Maaløe et al., 2019; Vahdat & Kautz, 2020) consisting of 1-D convolutional networks. For

the high-quality speech synthesis, BVAE-TTS learns mel-spectrogram distribution jointly with hierarchical latent variables in a bidirectional manner, where BVAE uses both bottom-up and top-down paths. Furthermore, to match the length of the target mel-spectrogram at inference, BVAE-TTS has an additional module called duration predictor, which predicts how many steps of a mel-spectrogram will be generated from each phoneme. To train the duration predictor, we employ an attention mechanism in BVAE-TTS to make BVAE-TTS utilize the text while learning attention maps between the text and the mel-spectrogram, where the mapping information is used for duration labels.

Our BVAE-TTS has advantages over the previous non-AR TTS models as follows:

- It has a simpler training process compared to the previous non-AR TTS models such as ParaNet (Peng et al., 2020) and FastSpeech (Ren et al., 2019). In the previous TTS models, well-trained AR teacher models are needed for duration labels or knowledge-distillation. Although FastSpeech 2 (Ren et al., 2020) removes the dependency on the teacher model, it still requires additional duration labels and acoustic features prepared in advance using other speech analysis methods. In contrast, BVAE-TTS requires only the text-speech paired dataset without any helps from the teacher model.

- It is more flexible in designing its architecture compared to the previous flow-based non-AR TTS models such as Flow-TTS (Miao et al., 2020) and Glow-TTS (Kim et al., 2020). The flow-based models have architectural constraints caused by their bijective transformation property, which leads to deeper models with a lot of parameters. On the contrary, the VAE-based model is free from the architectural constraints.

In experiments, we compare our BVAE-TTS with Tacotron 2 and Glow-TTS in terms of speech quality, inference speed, and model size. The results show that our model achieves 27 times speed improvement over Tacotron 2 in generating a mel-spectrogram with similar speech quality. Furthermore, BVAE-TTS outperforms the state-of-the-art non-AR TTS model, Glow-TTS, in both speech quality and inference time, while having 58% fewer model parameters. Additionally, we analyze how the latent representations are learned by BVAE-TTS. In this analysis, we confirm that the bottom part of BVAE-TTS captures the variation of mel-spectrograms that can occur from a text.

**Related work:** In the meantime, several TTS systems have utilized VAE to relax the one-to-many mapping nature in TTS, so improve the naturalness and the controllability of the systems. For example, (Hsu et al., 2018) and (Zhang et al., 2019) incorporate VAE to Tacotron 2 to learn the style or prosody of the input speech. However, previous uses of VAE have been limited to an auxiliary network in TTS based on the main AR TTS model. To the best of our knowledge, our BVAE-TTS is the first parallel TTS model that directly uses the VAE architecture to the task of TTS.

More discussions about other related works on the previous non-AR TTS models are in Section 5.

## 2 BACKGROUND

### 2.1 BIDIRECTIONAL-INFERENCE VARIATIONAL AUTOENCODER

Variational autoencoder (VAE) is a neural network generative model $p_\theta(\mathbf{x}, \mathbf{z})$ parameterized by $\theta$, where $\mathbf{x}$ is an observed data and $\mathbf{z}$ is a latent vector. In practice, since we only have a dataset $X = \{\mathbf{x_1}, ..., \mathbf{x_N}\}$ without the knowledge about $\mathbf{z}$, $\theta$ is typically optimized by maximizing the likelihood:

$$\max_\theta \log p_\theta(X) = \max_\theta \sum_{i=1}^{N} \log \int_{\mathbf{z}} p_\theta(\mathbf{x}_i, \mathbf{z}) d\mathbf{z}. \quad (1)$$

However, the integral over $\mathbf{z}$ is intractable to compute. Therefore, the VAE introduces an approximate posterior $q_\phi(\mathbf{z}|\mathbf{x})$ and does variational inference while maximizing the evidence lower bound (ELBO):

$$\log p_\theta(\mathbf{x}) \geq \mathbb{E}_{q_\phi(\mathbf{z}|\mathbf{x})} \left[ \log p_\theta(\mathbf{x}|\mathbf{z}) \right] - D_{KL} \left[ q_\phi(\mathbf{z}|\mathbf{x}) || p(\mathbf{z}) \right]. \quad (2)$$

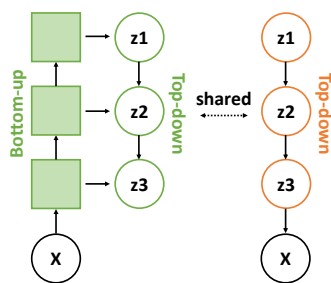

Figure 1: A Schematic diagram of the bidirectional-inference variational autoencoder. Samplings of latent variables occur in the layers expressed as circles.

In practice, for easy sampling and easy computation of the KL-divergence, each of the prior $p(\mathbf{z})$ and the approximate posterior $q_\phi(\mathbf{z}|\mathbf{x})$ is usually modeled as a multivariate normal distribution with a diagonal covariance matrix.

For a more expressive model, the latent vector $\mathbf{z}$ can be factorized into $\{\mathbf{z_1}, ..., \mathbf{z}_K\}$ with hierarchical dependency, where $K$ is the number of hierarchies. Then, each of the prior and the approximate posterior is represented as $p_\theta(\mathbf{z}) = \Pi_k p_\theta(\mathbf{z}_k|\mathbf{z}_{<k})$ and $q_\phi(\mathbf{z}|\mathbf{x}) = \Pi_k q_\phi(\mathbf{z}_k|\mathbf{z}_{<k}, \mathbf{x})$, respectively. In (Sønderby et al., 2016; Kingma et al., 2016; Vahdat & Kautz, 2020), the variational inference is designed in a bidirectional way based on bottom-up path and top-down path, while letting the inference network (left) and generative network (right) share their parameters as shown in Figure 1. First, along the bottom-up path, BVAE extracts hierarchical features from $\mathbf{x}$ and stores them inside of it. Then, along the top-down path, BVAE does the variational inference and reconstructs the input data considering the stored hierarchical features together. This architecture helps the model effectively learn the hierarchies between the latent variables, and equation (2) is changed as follows:

$$\log p_\theta(\mathbf{x}) \geq \mathbb{E}_{q_\phi(\mathbf{z}|\mathbf{x})}\left[\log p_\theta(\mathbf{x}|\mathbf{z})\right] - \sum_{k=1}^{K}\mathbb{E}_{q_\phi(\mathbf{z}_{<k}|\mathbf{x})}[D_{KL}\left[q_\phi(\mathbf{z}_k|\mathbf{x}, \mathbf{z}_{<k})||p(\mathbf{z}_k|\mathbf{z}_{<k})]\right]. \quad (3)$$

## 2.2 DURATION PREDICTOR IN NON-AUTOREGRESSIVE TEXT-TO-SPEECH

To achieve the non-autoregressive (non-AR) text-to-speech (TTS) model, the model needs to predict the length of the target mel-spectrogram from a text. This is because there is no way to access to the length of the target mel-spectrogram at inference. However, this is a challenging task considering that there are no pre-defined rules between the lengths of text and mel-spectrogram. Recently, several non-AR TTS models (Ren et al., 2019; Zeng et al., 2020; Kim et al., 2020) resolved the issue by introducing a module called duration predictor. The duration predictor is a module that predicts how many mel-spectrogram steps will be generated from each phoneme.

First, using the duration predictor, the non-AR TTS models compute durations $\hat{D} = \{\hat{d}_1, ..., \hat{d}_S\}$ corresponding to each phoneme based on phoneme representations $H = \{\mathbf{h}_1, ..., \mathbf{h}_S\}$, where each $\hat{d}_i$ is a positive integer that is rounded off from a positive real number, and $S$ is the number of phonemes. Then, $H$ is expanded to the length of the target mel-spectrogram $T$, by repeating each $\mathbf{h}_i$ as many steps as $\hat{d}_i$. Finally, the non-AR TTS models generate a mel-spectrogram in parallel by decoding the expanded phoneme representations.

In practice, since there are no ground-truth duration labels for the training of the duration predictor, the non-AR models obtain the duration labels using various methods, and we adopt a method used in FastSpeech (Ren et al., 2019). From well-aligned attention maps, the duration labels are obtained according to $d_i = \sum_{t=1}^{t=T}[\text{argmax}_s\ a_{s,t} == i]$, where $a_{s,t}$ represents an attention weight given from the $t$-th mel-spectrogram step to the $s$-th phoneme.

## 3 METHODOLOGY

In this section, we explain a novel non-autoregressive (non-AR) TTS model, BVAE-TTS, which is based on the bidirectional-inference variational autoencoder (BVAE). As shown in Figure 2-(a), during training, BVAE-TTS is given a mel-spectrogram with a phoneme sequence, and it is trained to reconstruct the mel-spectrogram while maximizing the ELBO. Here, the duration predictor is jointly trained using the attention maps that BVAE-TTS generates during training. As shown in Figure 2-(c), at inference BVAE-TTS generates a mel-spectrogram from a phoneme sequence using the duration predictor as described in Section 2.2, while using its top-down path for decoding the expanded phoneme representations. In Appendix A.1, pseudo-codes for the training and inference of BVAE-TTS are contained for detailed descriptions. The other aspects of BVAE-TTS are described in the following sub-sections in more detail.

### 3.1 USING BVAE FOR TEXT-TO-SPEECH

Unlike the previous BVAE models (Sønderby et al., 2016; Kingma et al., 2016; Vahdat & Kautz, 2020) are trained to generate natural images, our model should learn to generate a mel-spectrogram

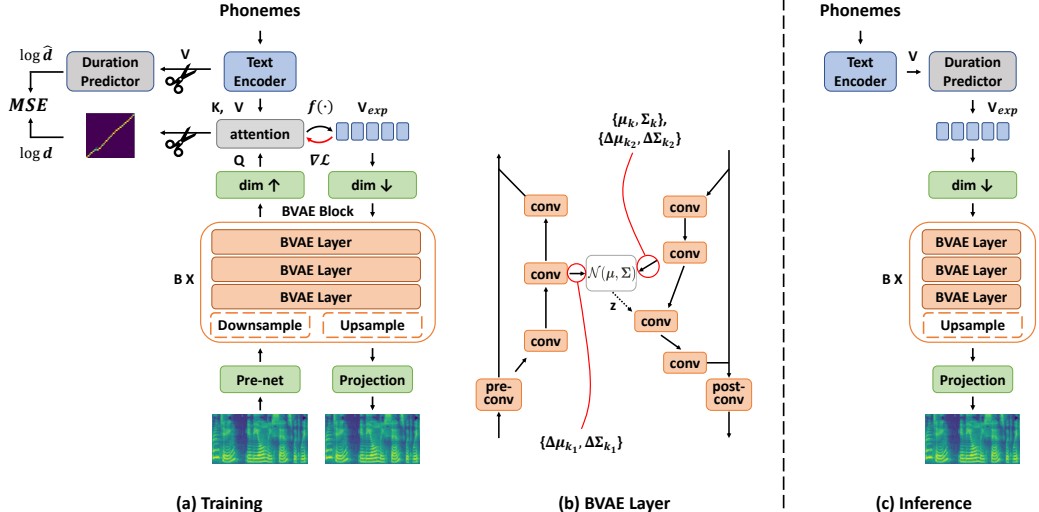

**(a) Training**  **(b) BVAE Layer**  **(c) Inference**

Figure 2: (a) The training procedure of BVAE-TTS. The scissors represent to detach the signal from the computational graph to block the gradient signal in backpropagation. The Downsample and Upsample layers are included in even-numbered BVAE blocks. $\mathbf{V_{exp}}$ represents the expanded $\mathbf{V}$ to fit the length of the top-down path input. The $f(\cdot)$ represents the Straight-Through argmax with jitter, and $\nabla\mathcal{L}$ represents the gradient signal. (b) BVAE Layer. The dotted arrow represents sampling. The red lines indicate the parameters of prior and approximate posterior normal distributions. (c) The inference procedure of BVAE-TTS that uses the top-down path only.

that is not only natural but also corresponding to the input text. To this end, we add a dot-product attention network (Bahdanau et al., 2015) on top of the BVAE, which is a channel for BVAE-TTS to learn how to utilize the text properly. First, using a text encoder, key ($\mathbf{K}$) and value ($\mathbf{V}$) are obtained from a phoneme sequence, and from the bottom-up path, query ($\mathbf{Q}$) is obtained. Here, obtaining $\mathbf{Q}$ is different from the bottom-up paths of the previous BVAE studies used in the image domain, where only the parameters for posterior approximation are obtained. Second, based on the dot-product attention with $\mathbf{Q}$, $\mathbf{K}$, and $\mathbf{V}$, the $\mathbf{V}$ are expanded to $\mathbf{V_{exp}}$ to fit the length of the top-down path, and then the $\mathbf{V_{exp}}$ is inputted into the top-down path of BVAE-TTS. Lastly, the BVAE-TTS does both the variational inference and mel-spectrogram reconstruction along the top-down path using the expanded text representations with the following objectives:

$$\mathcal{L}_{recon} = -\mathbb{E}_{q_\phi(\mathbf{z}|\mathbf{x},\mathbf{y})}\left[\log p_\theta(\mathbf{x}|\mathbf{z},\mathbf{y})\right], \tag{4}$$

$$\mathcal{L}_{KL} = \sum_{k=1}^{K}\mathbb{E}_{q_\phi(\mathbf{z}_{<k}|\mathbf{x},\mathbf{y})}\left[D_{KL}\left[q_\phi(\mathbf{z}_k|\mathbf{x},\mathbf{z}_{<k},\mathbf{y})||p(\mathbf{z}_k|\mathbf{z}_{<k},\mathbf{y})\right]\right], \tag{5}$$

where $\mathbf{x}$ represents mel-spectrogram, $\mathbf{y}$ represents text, $\mathbf{z}$ represents latent representation, and mean absolute error (MAE) loss is used for the $\mathcal{L}_{recon}$.

In addition to that, a duration predictor is jointly trained to predict durations corresponding to each phoneme in the logarithmic domain using mean square error (MSE) loss, $\mathcal{L}_{dur} = \mathbb{E}[(\log d_i - \log \hat{d}_i)^2]$, where $d_i$ and $\hat{d}_i$ are obtained as described in Section 2.2. The duration predictor takes as input the $\mathbf{V}$ obtained from the text encoder, and here the $\mathbf{V}$ is detached from the computational graph to prevent it from affecting the BVAE training.

## 3.2 ARCHITECTURE OF BVAE-TTS

In this section, we describe the architecture of BVAE-TTS that hierarchically learns the latent representations based on BVAE blocks consisting of BVAE layers.

**BVAE block:** As shown in Figure 2-(a), the main part of BVAE-TTS is the stacked BVAE blocks, each consisting of BVAE layers. To guide the multi-scale fine-to-coarse latent features to be contained in the latent hierarchies, the time dimension of input mel-spectrogram is downsampled using bilinear downsampling operations (Zhang, 2019) in even-numbered BVAE blocks along the bottom-up path. Here, the numbering of BVAE blocks starts with one and increases from bottom to top. On the contrary, its time dimension is upsampled again along the top-down path, by repeating the signals in the BVAE blocks where the downsamplings have occurred. In odd-numbered BVAE-blocks, the channel dimension is decreased along the bottom-up path and is increased along the top-down path. It is done at the pre- or post-convolutional network of the first BVAE layer in the BVAE blocks shown in Figure 2-(b).

**BVAE layer:** The main element of the BVAE block is the BVAE layer. As shown in Figure 2-(b), at each bottom-up and top-down path, the parameters of the prior and the approximate posterior distributions $\{\boldsymbol{\mu}_k, \boldsymbol{\Sigma}_k\}$, $\{\Delta\boldsymbol{\mu}_{k_1}, \Delta\boldsymbol{\Sigma}_{k_1}\}$, $\{\Delta\boldsymbol{\mu}_{k_2}, \Delta\boldsymbol{\Sigma}_{k_2}\}$ are obtained from 1-D convolutional networks. Then, the prior distribution $p_\theta(\mathbf{z}_k|\mathbf{z}_{<k}, \mathbf{y})$, and the approximate posterior distribution $q_\phi(\mathbf{z}_k|\mathbf{z}_{<k}, \mathbf{x}, \mathbf{y})$ are defined as follow:

$$p_\theta(\mathbf{z}_k|\mathbf{z}_{<k}, \mathbf{y}) := \mathcal{N}(\boldsymbol{\mu}_k, \ \boldsymbol{\Sigma}_k), \tag{6}$$

$$q_\phi(\mathbf{z}_k|\mathbf{z}_{<k}, \mathbf{x}, \mathbf{y}) := \mathcal{N}(\boldsymbol{\mu}_k + \Delta\boldsymbol{\mu}_{k_1} + \Delta\boldsymbol{\mu}_{k_2}, \ \boldsymbol{\Sigma}_k \cdot \Delta\boldsymbol{\Sigma}_{k_1} \cdot \Delta\boldsymbol{\Sigma}_{k_2}), \tag{7}$$

where diagonal covariance matrices $\boldsymbol{\Sigma}$ are used after applying a softplus function to guarantee that they are positive. This parameterization follows (Vahdat & Kautz, 2020), where the approximate posterior $q_\phi(\mathbf{z}_k|\mathbf{z}_{<k}, \mathbf{x}, \mathbf{y})$ is relative to the prior $p_\theta(\mathbf{z}_k|\mathbf{z}_{<k}, \mathbf{y})$. With this parameterization, when the prior moves, the approximate posterior moves accordingly while making the BVAE training easier and more stable. During training, the latent representation $\mathbf{z}_k$ is sampled from $q_\phi(\mathbf{z}_k|\mathbf{z}_{<k}, \mathbf{x}, \mathbf{y})$, and sampled from $p_\theta(\mathbf{z}_k|\mathbf{z}_{<k}, \mathbf{y})$ at inference. Other details on the BVAE-TTS architecture such as text encoder or duration predictor are in Appendix A.2.

## 3.3 BRIDGE THE GAP BETWEEN TRAINING AND INFERENCE

When BVAE-TTS reconstructs a mel-spectrogram during training, text representations are expanded via the attention network. In contrast, text representations are expanded via the duration predictor at inference. Therefore, to bridge the gap between the attention-based mel-spectrogram generation and the duration-based mel-spectrogram generation, we use the following techniques in this work.

**Straight-Through argmax:** In the duration-based generation, the predicted duration of each phoneme is used after it is rounded to the nearest integer. It means that there is a corresponding phoneme for every time step of a mel-spectrogram. Therefore, during training, we use a trick called Straight-Through (ST) argmax, where the phoneme representation given the largest attention weight from each query time step, which is computed using $\arg\max$ operation, is passed to the top-down path instead of the weighted sum in the attention mechanism. However, during backpropagation, the parameter update is conducted as if the signal was a result of the weighted sum.

**Jitter:** To make the model more robust to the errors of the duration predictor, we apply jitter to the text representations, where each of the text representations obtained from the ST-argmax is replaced with a text representation attended by one of the neighboring queries with each probability of 25% during training. We also experimentally observe that applying jitter makes the learning of the attention maps more stable, so the attention maps are not defused throughout the training and stay diagonal.

**Positional encoding biasing & Guided attention:** In order to reduce the gap between the attention-based generation and the duration-based generation, it is important for the learned attention maps to have diagonal shapes. Therefore, we use two additional techniques to directly help BVAE-TTS learn the diagonal attention maps. First, we add positional encoding vectors with different angular speeds to query and key as an inductive bias following (Ping et al., 2018). Second, we use an additional guided attention loss $\mathcal{L}_{guide}$ that gives penalties for attention weights deviating from the diagonal following (Tachibana et al., 2018). For more details on the techniques in this section, see Appendix A.3.

With the above techniques, BVAE-TTS is trained with the following objective:

$$\mathcal{L}_{total} = \mathcal{L}_{recon} + \alpha * \mathcal{L}_{KL} + \mathcal{L}_{dur} + \mathcal{L}_{guide}, \tag{8}$$

where the $\alpha$ is a warm-up constant that linearly increases from 0 to 1 over the first 20% of training. This technique is proposed in (Sønderby et al., 2016) to weaken the variational regularization in the early stages of training.

# 4 EXPERIMENTS

In this section, we describe the experimental setup and the results obtained from the quantitative and qualitative experiments that are conducted to evaluate our BVAE-TTS[1]. For comparison, we use two state-of-the-art TTS models: Tacotron 2[2] for an AR TTS model and Glow-TTS[3] for a non-AR TTS model. Here, we use the pre-trained weights of the models that are publicly available.

## 4.1 EXPERIMENTAL SETUP

In the experiments, we mainly use the LJSpeech dataset (Ito & Johnson, 2017) consisting of 12500 / 100 / 500 samples for training / validation / test splits, respectively. For speech data, we convert raw waveforms into log-mel-spectrograms with 1024 window length and 256 hop length and use them as target sequences of our BVAE-TTS model. For text data, we convert raw texts into phoneme sequences using grapheme-to-phoneme library (Park, 2019) and use them as input sequences of BVAE-TTS.

We train the BVAE-TTS consisting of 4 BVAE blocks for 300K iterations with a batch size of 128. For an optimizer, we use the Adamax Optimizer (Kingma & Ba, 2015) with $\beta_1 = 0.9$, $\beta_2 = 0.999$ using the learning rate scheduling used in (Vaswani et al., 2017), where initial learning rate of 1e-3 and warm-up step of 4000 are used. Training of BVAE-TTS takes about 48 hours on Intel(R) Xeon(R) Gold 5120 CPU (2.2GHz) and NVIDIA V100 GPU on the Pytorch 1.16.0 library with Python 3.6.10 over the Ubuntu 16.04 LTS. For more details on the hyperparameters, see Appendix A.4.

## 4.2 EXPERIMENTAL RESULTS

In this section, we compare BVAE-TTS with Tacotron 2 and Glow-TTS in terms of speech quality, inference time, and model size. For the quality evaluation, we use pre-trained WaveGlow[4] as a vocoder that converts mel-spectrograms to waveforms. When we sample latent representations in Glow-TTS and BVAE-TTS, we use the temperature of 0.333 for the models for better speech quality. (Kingma & Dhariwal, 2018)

Table 1: Experimental results. The MOS-ID and MOS-OOD are written with 95% confidence intervals. The number in parentheses represents the number of parameters of BVAE-TTS that are used at inference.

| Method | MOS-ID | MOS-OOD | Inference Time (ms) | # of Parameters |
|---|---|---|---|---|
| GT Audio | $4.68 \pm 0.06$ | - | - | - |
| GT Mel-spectrogram | $4.41 \pm 0.07$ | - | - | - |
| Tacotron 2 | $4.35 \pm 0.07$ | $4.16 \pm 0.07$ | 658.5 | 28.2M |
| Glow-TTS | $3.96 \pm 0.08$ | $3.89 \pm 0.10$ | 43.07 | 28.6M |
| BVAE-TTS | $4.14 \pm 0.07$ | $4.21 \pm 0.07$ | 24.20 | 16.0M (12.0M) |

**Speech quality:** In this experiment, we measure the Mean-Opinion-Score (MOS) for audio samples generated by each TTS model using fifty sentences randomly sampled from the in-domain LJSpeech test set (MOS-ID). In addition, we measure another MOS on fifty sentences randomly sampled from the test-clean set of LibriTTS (Zen et al., 2019) to see the generalization ability on the out-of-domain

---

[1]https://github.com/LEEYOONHYUNG/BVAE-TTS

[2]https://github.com/NVIDIA/tacotron2

[3]https://github.com/jaywalnut310/glow-tts

[4]http://github.com/NVIDIA/waveglow

text data (MOS-OOD). Via Amazon Mechanical Turk (AMT), we assign five testers living in the United States to each audio sample, and ask them to listen to the audio sample and give a score between 1 to 5 in 9-scale based on its naturalness.

The MOS results shown in Table 1 demonstrate the superiority of our BVAE-TTS, where it out-performs the state-of-the-art non-AR TTS model, Glow-TTS, on both MOS-ID and MOS-OOD. Although BVAE-TTS does not surpass Tacotron 2 in MOS-ID, our model achieves better results on MOS-OOD. It shows its robustness to the out-of-domain text over the autoregressive TTS model which suffers from the accumulated prediction error. For a better understanding of the speech quality generated by the models, we strongly encourage the readers to listen to the audio samples in the supplementary material[5] or on the demo page[6].

**Inference time:** We measure inference times taken to generate a mel-spectrogram from a text on the 500 sentences of LJSpeech test set in GPU environment. The average inference time of each TTS model is shown in Table 1. As can be seen in the table, our BVAE-TTS shows 27.2 times faster inference speed on average compared with Tacotron 2, and it is also 1.78 times faster than Glow-TTS. Besides, due to the sequential generation property of the AR TTS model, the gap between the inference speed of BVAE-TTS and Tacotron 2 increases with a longer length of an input text. See Appendix B for more details.

**Model size:** As can be seen in the last column of Table 1, BVAE-TTS has the smallest number of parameters of 16.0M while maintaining high-quality speech synthesis. Furthermore, BVAE-TTS gets smaller (to 12.0M) at inference because the layers belonging to the bottom-up path are not used to generate a mel-spectrogram, where the number of parameters is 58% fewer parameters compared to Glow-TTS. This shows that the training principle of BVAE-TTS that hierarchically learns the latent features while adjusting hidden dimensions enables our model to have small parameters. It is contrary to the flow-based TTS models such as Flow-TTS (Miao et al., 2020) and Glow-TTS (Kim et al., 2020), where many parameters are used due to its architectural constraint.

### 4.3 MODEL ANALYSIS

As our BVAE-TTS is the first VAE-based parallel TTS model, we conduct several analyses on it. First, we analyze BVAE-TTS to see how the hierarchies are contained in the latent representations and how the variance in mel-spectrograms is learned. Then, we verify the effectiveness of the techniques used in BVAE-TTS such as Straight-Through (ST) argmax and jitter through ablation studies.

#### 4.3.1 ANALYSIS ON HIERARCHY

In this experiment, we conduct an analysis on the hierarchical latent representation learn-ing of BVAE-TTS. To see how the latent fea-tures of the mel-spectrograms are learned in the hierarchies, we observe the variations of the mel-spectrograms sampled from the same text, while using different temperatures for dif-ferent BVAE blocks. Specifically, we set a tar-get BVAE block among the four BVAE blocks and increase the variance of the target BVAE block, by using a temperature of 1.0 or 2.0 or 5.0 for the samplings occured in the BVAE lay-ers belonging to the target BVAE block. On the contrary, we lower the variance of the non-target BVAE blocks using a temperature of 0.333. Then, we sample 100 different mel-spectrograms each from the same text, while varying the target BVAE block and its tempera-ture.

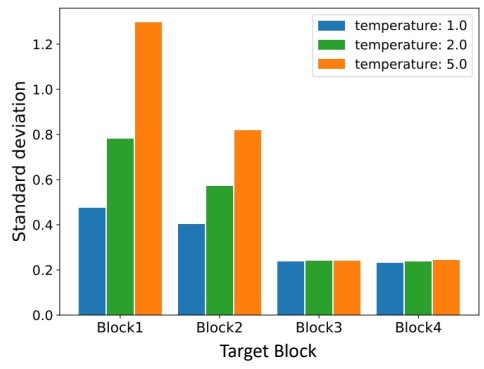

Figure 3: Averages of pixel-by-pixel standard de-viations measured on randomly sampled 100 mel-spectrograms.

---

[5]https://openreview.net/forum?id=o3iritJHLfO

[6]https://leeyoonhyung.github.io/BVAE-TTS

Figure 3 shows the averages of pixel-by-pixel standard deviations measured on the randomly sampled 100 mel-spectrograms from the same text. The block numbers in the figure are given from one to four starting from the BVAE block at the bottom. In this experiment, we observe that the variance of the speech is mostly contained in the latent representations of BVAE blocks 1, 2, which are close to the mel-spectrogram. However, there is not much variance in the generated mel-spectrograms when we increase the temperature of BVAE blocks 3, 4, which are close to the text representations. Therefore, we can conclude that the global content is mostly contained in the expanded text representations obtained using the text encoder and the duration predictor, and the BVAE blocks 3, 4 focus on building the content rather than its style. Note that while Figure 3 shows standard deviations measured using one examplar sentence, "One, Two, Three.", the tendency is consistent regardless of the input sentence. Mel-spectrogram samples obtained in this experiment are in Appendix C.

### 4.3.2 ABLATION STUDY

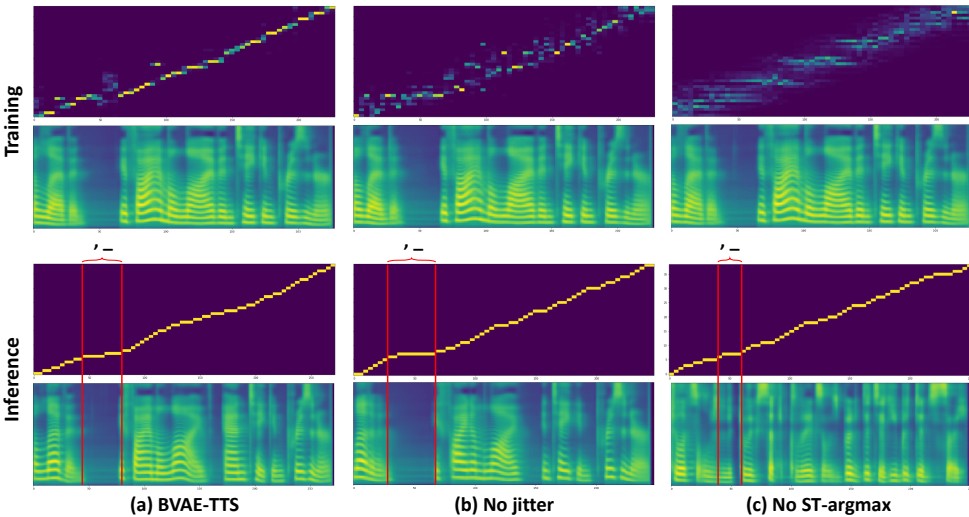

Figure 4: Examples of ablations. Each row represents (1) learned attention map, (2) reconstructed mel-spectrogram, (3) diagonally stacked predicted durations, (4) generated mel-spectrogram from a text. (1), (2) are obtained during training, and (3),(4) are obtained at inference. The red lines represent the section of the durations corresponding to a 'comma' and a 'whitespace' in text.

We conduct ablation studies to see the effects of applying jitter and Straight-Through (ST) argmax to the soft attention mechanism in BVAE-TTS, and the results are shown in Figure 4. Here, since jitter is included in ST-argmax (jitter is applied to the output of the $\arg\max$), the ablation study of not using ST-argmax represents when BVAE-TTS is trained using a normal soft attention mechanism.

The most noticeable differences appear in the attention maps that they learn. As shown in the first row of Figure 4-(a), (b), applying jitter shows the effectiveness for BVAE-TTS to learn well-aligned attention maps. It results in using more accurate duration labels to train the duration predictor, which leads to more natural speech. We observe that BVAE-TTS without applying jitter still generates a clear speech even though it is a little unnatural, where it obtains a 3.68 MOS on the LJSpeech dataset. As shown in the bottom mel-spectrogram of Figure 4-(c), the BVAE-TTS without ST-argmax technique just generates a stuttering sound.

As shown in Figure 4-(a), although the BVAE-TTS also does not learn the perfect attention map, BVAE-TTS successfully generates a mel-spectrogram at inference. Since the text is forced to be used monotonically in the duration-based generation, it makes the model more robust to the attention errors while making fewer pronouncing mistakes. In addition, when using the duration predictor, it is also possible to locally control the speed of speech by adjusting the predicted durations. The experiment on the speed control is included in Appendix D.

## 5 DISCUSSION

To overcome the limitations that the autoregressive (AR) TTS models have, various non-AR architectures have been recently proposed. On one hand, there have been feed-forward neural networks such as ParaNet (Peng et al., 2020), and FastSpeech 1, 2 (Ren et al., 2019; 2020), which use knowledge distillation or additional duration labels and acoustic features. Although they succeeded in enabling their models to predict the lengths of the target mel-spectrograms, the feed-forward architectures did not fit the one-to-many mapping problems in TTS. Therefore, FastSpeech (Ren et al., 2019) uses as targets mel-spectrograms that the AR teacher model generates. This is because much of the diversity in original mel-spectrograms has been eliminated in the mel-spectrograms generated by the AR teacher model. Besides, FastSpeech 2 (Ren et al., 2020) even directly uses additional pre-obtained acoustic features such as pitch and energy to relax the one-to-many mapping nature in TTS. In contrast to the models, it can be seen that BVAE-TTS is asked to solve one-to-one mapping problems during training because there is only one possible target for the reconstruction task. As a result, BVAE-TTS can generate natural and diverse samples while learning latent features in mel-spectrograms.

On the other hand, there have been generative flow-based non-AR TTS models such as Flow-TTS (Miao et al., 2020) and Glow-TTS (Kim et al., 2020). While their speech quality is comparable to that of AR TTS models, flow-based generative models usually have a problem that they require a lot of model parameters. In the models, the dimensions of the hidden representations in the flow networks should be the same through the whole network, and their bipartite flow requires many layers and larger hidden size because of its lack of expressiveness (Ping et al., 2019; Lee et al., 2020). Contrary to flow-based TTS models, our BVAE-TTS is free from the above issue. In this work, by designing BVAE-TTS in hierarchical architecture with varying hidden dimensions, we can outperform the flow-based TTS model, Glow-TTS in both speech quality and speed, while having a much smaller model size.

## 6 CONCLUSION

In this work, we propose BVAE-TTS, which is the first VAE-based non-AR TTS model that generates a mel-spectrogram from a text in parallel. To use the BVAE architecture in text-to-speech, we combine BVAE with an attention mechanism to utilize a text, and to extract duration labels for the training of the duration predictor. In experiments, BVAE-TTS achieves to generate speech 27x faster than Tacotron 2 with similar speech quality, and also outperforms Glow-TTS in terms of both speech quality and inference time with 58% fewer parameters. Since our VAE-based TTS model shows competitive performance and has many advantages over the previous non-AR TTS models, we hope it becomes a good starting point for future VAE-based TTS research.

ACKNOWLEDGMENTS

K. Jung is with ASRI, Seoul National University, Korea. This work was supported by the Ministry of Trade, Industry & Energy (MOTIE, Korea) under the Industrial Technology Innovation Program (No. 10073144). This research was results of a study on the "HPC Support" Project, supported by the 'Ministry of Science and ICT' and NIPA.

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

# A DETAILS ON BVAE-TTS

## A.1 ALGORITHMS

---

**Algorithm 1:** Pseudo-code of BVAE-TTS training

---

**Data:**

  **x, y**: a mel-spectrogram, a phoneme sequence

  **Q, K, V**, $\mathbf{V_{exp}}$: query, key, value, and expanded value matrices

  $h, h_{exp}$: hidden representations, expanded hidden representations

  BVAE-TTS$[b, l]$: an $l$-th BVAE layer in $b$-th BVAE block.

  $PE_{query}, PE_{key}$: positional encoding vectors for query and key

  **A**: a soft attention map obtained before applying the ST-argmax technique

  $D$: phoneme durations extracted from the attention map

  $\hat{D}$: phoneme durations predicted by the duration predictor

  $\alpha$: a warm-up constant

**Result:**

  $\hat{\mathbf{x}}$: a reconstructed mel-spectrogram

  $\mathcal{L}_{total}, \mathcal{L}_{recon}, \mathcal{L}_{KL}, \mathcal{L}_{dur}, \mathcal{L}_{guide}$: total loss, and losses that make up the total loss

**K, V** ← `TextEncoder`(**y**);

$h \leftarrow$ `PreNet`(**x**)

**for** $b \leftarrow 0$ **to** $B - 1$ **do**

  **if** $b\%2 == 1$ **then**

    $h \leftarrow$ `Downsample`($h$);

  **end**

  **for** $l \leftarrow 0$ **to** $L - 1$ **do**

    $h, \Delta\boldsymbol{\mu}_{k_1}, \Delta\boldsymbol{\Sigma}_{k_1} \leftarrow$ BVAE-TTS$[b, l]$.`BottomUp`($h$);

    BVAE-TTS$[b, l].\Delta\boldsymbol{\mu}_{k_1} \leftarrow \Delta\boldsymbol{\mu}_{k_1}$;

    BVAE-TTS$[b, l].\Delta\boldsymbol{\Sigma}_{k_1} \leftarrow$ `Softplus`($\Delta\boldsymbol{\Sigma}_{k_1}$);

  **end**

**end**

$\mathbf{Q} \leftarrow$`IncreaseDimension`($h$);

$\mathbf{V_{exp}}, \mathbf{A} \leftarrow$ `Attention`($\boldsymbol{Q} + PE_{query}, \ \boldsymbol{K} + PE_{key}, \ \boldsymbol{V}$);

$h_{exp} \leftarrow$`DecreaseDimension`($\mathbf{V_{exp}}$);

$\mathcal{L}_{KL} \leftarrow 0$;

**for** $b \leftarrow B - 1$ **to** $0$ **do**

  **for** $l \leftarrow L - 1$ **to** $0$ **do**

    $h_{exp}, kl\_loss \leftarrow$ BVAE-TTS$[b, l]$.`TopDown`($h_{exp}$);

    $\mathcal{L}_{KL} \leftarrow \mathcal{L}_{KL} + kl\_loss$

  **end**

  **if** $b\%2 == 1$ **then**

    $h_{exp} \leftarrow$ `Upsample`($h_{exp}$);

  **end**

**end**

$\hat{\mathbf{x}} \leftarrow$ `Projection`($h_{exp}$);

$D \leftarrow$ `ExtractDurations`($\boldsymbol{A}$);

$\hat{D} \leftarrow$ `DurationPredictor`($\boldsymbol{V}$);

$\mathcal{L}_{recon} \leftarrow$ `MAE`($\mathbf{x}, \hat{\mathbf{x}}$);

$\mathcal{L}_{dur} \leftarrow$ `MSE`($\log(D), \log(\hat{D})$);

$\mathcal{L}_{guide} \leftarrow$ `GuidedAttentionLoss`($A$);

$\mathcal{L}_{total} \leftarrow \mathcal{L}_{recon} + \alpha \cdot \mathcal{L}_{KL} + \mathcal{L}_{dur} + \mathcal{L}_{guide}$;

---

---

**Algorithm 2:** Pseudo-code of BVAE-TTS inference

**Data:**

    **x, y**: a mel-spectrogram, a phoneme sequence

    **K, V**, $\mathbf{V_{exp}}$: key, value, and expanded value matrices

    $h_{exp}$: expanded hidden representations

    BVAE-TTS$[b, l]$: an $l$-th BVAE layer in $b$-th BVAE block.

    $\hat{D}$: phoneme durations predicted by the duration predictor

**Result:**

    $\hat{\mathbf{x}}$: a reconstructed mel-spectrogram

$\mathbf{K}, \mathbf{V} \leftarrow$ `TextEncoder`$(\mathbf{y})$;

$\hat{D} \leftarrow$ `DurationPredictor`$(\mathbf{V})$;

$\mathbf{V_{exp}} \leftarrow$ `ExpandRepresentations`$(\mathbf{V}, \hat{D})$;

$h_{exp} \leftarrow$ `DecreaseDimension`$(\mathbf{V_{exp}})$;

**for** $b \leftarrow B - 1$ **to** 0 **do**

    **for** $l \leftarrow L - 1$ **to** *0* **do**

        $h_{exp, \_} \leftarrow$ BVAE-TTS$[b, l]$.`TopDown`$(h_{exp})$;

    **end**

    **if** $b\%2 == 1$ **then**

        $h_{exp} \leftarrow$ `Upsample`$(h_{exp})$;

    **end**

**end**

$\hat{\mathbf{x}} \leftarrow$ `Projection`$(h_{exp})$;

---

## A.2 ARCHITECTURE DETAILS

**Text encoder:** To obtain key (**K**) and value (**V**) used in the attention mechanism, we use a text encoder used in DeepVoice 3 (Ping et al., 2018), which consists of a series of 1-D convolutional network with a gated linear unit and a residual connection; $y = \sqrt{0.5}(x + \text{conv}_1(x) * \sigma(\text{conv}_2(x)))$. The last hidden representations of the text encoder are used as **K**, and **V** is computed according to $\mathbf{V} = \sqrt{0.5}(\mathbf{K} + \mathbf{E})$, where **E** represents phoneme embedding vectors.

**Duration predictor:** We use a duration predictor used in FastSpeech (Ren et al., 2019), which consists of two Conv1d-ELU-LayerNorm-Dropout blocks, followed by a linear projection layer. Then, we take exponential of the output values and add ones to them to guarantee that the predicted durations are larger than one.

**Pre-net & Projection layer & Dimension matching layers:** To adjust dimensions properly, there are four additional layers; Pre-net, Projection layer, and two additional dimension matching layers. Pre-net consists of two Conv1d-Dropout-ELU blocks, and Projection layer consists of two Conv1d-ELU-LayerNorm-Dropout blocks followed by a Conv1d layer with a sigmoid activation. For the dimension matching layers, we use a Conv1d layer with kernel width 5 to increase the dimension, and use a linear layer to decrease the dimension.

**BVAE block & layer:** To increase the receptive field of the model, the pre- & post- convolutional networks use dilated convolution with the dilation factors 1, 2, 4 for each BVAE layer in each BVAE block. Every convolutional network except for the networks that output the parameters of the prior and approximate posterior distributions are followed by an ELU layer. When the residual path is merged with the original path, we combine the signals according to $y = \sqrt{0.5}(x + f(x))$ to stabilize training. Following (Vahdat & Kautz, 2020), we apply the spectral normalization (Miyato et al., 2018) to the convolutional networks located on the residual path to stabilize training.

## A.3 ADDITIONAL TECHNIQUES

**Straight-Through argmax:** Let $\mathbf{A} = \{a_{s,t}\}$ be the attention matrix obtained from the dot-product operation in the attention mechanism. Then, the original attention mechanism expands the value **V**

to $\mathbf{V_{exp}}$ according to $\mathbf{V_{exp}} = \mathbf{A}^\intercal \cdot \mathbf{V}$. However, with the ST-argmax, another one-hot attention matrix $\mathbf{A_{onehot}} = \{a'_{s,t}\}$ is firstly obtained, where for $s$ and $t$, $a'_{s,t} = 1$ if $a_{s,t} == \max_i(a_{i,t})$ else 0. Then $\mathbf{V_{exp}}$ is obtained as follow:

$$\mathbf{V_{exp}} = (\mathbf{A} + (\mathbf{A_{onehot}} - \mathbf{A}).\text{detach}())^\intercal \mathbf{V}, \tag{9}$$

where by using the detach function, we let the gradient flow to $\mathbf{A}$ instead of $\mathbf{A_{onehot}}$ during back-propagation.

**Jitter:** To apply jitter to the hidden states, we apply jitter to $\mathbf{A_{onehot}}$ obtained from ST-argmax, of which example is shown in Figure 5. By using a function '**random.choices**' that returns an element chosen from a list according to probabilities $\mathbf{p}$, applying jitter to $\mathbf{A_{onehot}}$ can be implemented according to $a'_{s,t} = \textbf{random.choices}([a'_{s,t-1},\ a'_{s,t},\ a'_{s,t+1}],\ \mathbf{p} = [0.25, 0.5, 0.25])$.

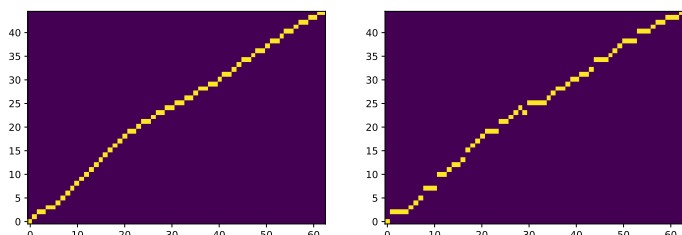

Figure 5: An example of applying jitter to ideal diagonal attention matrix $\mathbf{A_{onehot}}$

**Positional encoding biasing:** We add positional encoding vectors with different angular speed $w_s$ to query $\mathbf{Q}$ and key $\mathbf{K}$ to introduce inductive bias of diagonal attention:

$$PE_{(pos,2i)} = sin(\frac{w_s * pos}{10000^{2i/d}}), \tag{10}$$

$$PE_{(pos,2i+1)} = cos(\frac{w_s * pos}{10000^{2i/d}}), \tag{11}$$

where $pos$ is the position, $i$ is the dimension, and $d$ is the dimension of $\mathbf{Q}$ and $\mathbf{K}$.

For $\mathbf{Q}$, $w_s = 1$, and for $\mathbf{K}$, $w_s = T/S$, where $T$ and $S$ are the numbers of time steps of $\mathbf{Q}$ and $\mathbf{K}$. Then, the attention matrix $\mathbf{A}$ is obtained according to $\mathbf{A} = (K + PE_{key}) \cdot (Q + PE_{query})^\intercal$.

**Guided attention:** To directly lead the attention map to be formed in diagonal, we additionally introduce $\mathcal{L}_{guide}$ as follow:

$$\mathcal{L}_{guide} = \mathbb{E}[a_{s,t} * (1 - \exp(-(s/S - t/T)^2)/2g^2)],$$

where $g$ is set to 0.2.

## A.4 HYPER PARAMETERS

Table 2: Hyperparameters of BVAE-TTS.

| Hyperparameter | BVAE-TTS |
|---|---|
| Phoneme Embedding Dimension | 256 |
| Text Encoder Layers | 7 |
| Text Encoder Hidden Dimension | 256 |
| Text Encoder Conv1D Kernel Width | 5 |
| Text Encoder Conv1D Filter Size | 256 |
| Text Encoder Dropout | 0.1 |
| Pre-net Layers | 2 |
| Pre-net Dropout | 0.5 |
| Pre-net Hidden Dimension | 256 |
| Downsampling Conv 1D kernel | [0.25, 0.5, 0.25] |
| Projection Layers | 3 |
| Projection Dropout | 0.5 |
| Projection Conv1D Kernel Width | 5 |
| Projection Conv1D Filter Size | 256 |
| Duration Predictor Conv1D Kernel Width | 3 |
| Duration Predictor Conv1D Filter Size | 256 |
| Duration Predictor Dropout | 0.1 |
| BVAE Blocks | 4 |
| BVAE Layers per block | 3 |
| BVAE Conv1D Kernel Width | 5 |
| Hidden Dimensions of BVAE blocks | 128, 128, 64, 64 |
| **Total Number of Parameters** | **16.0M (12.0M)** |

## B PARALLEL SYNTHESIS

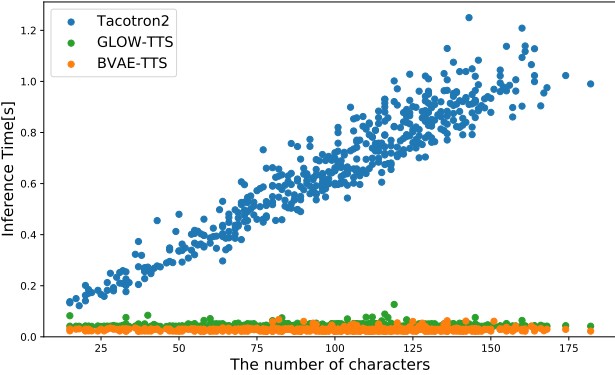

Figure 6: Inference time measured on 500 sentences of LJSpeech test set.

To see the benefit of parallel mel-spectrogram synthesis, we measure the inference time of Tacotron 2, Glow-TTS, and BVAE-TTS on the 500 sentences of LJSpeech test set in GPU environment. Figure 6 shows that the inference times of the non-AR TTS models are almost constant even if the length of the input text gets longer. On the contrary, the inference time of Tacotron2 linearly increases.

## C ANALYSIS ON HIERARCHY

While changing the target BVAE block and its temperature as described in 4.3.1, we observe the generated mel-spectrogram samples. As shown in Figure 7-9, the variance of the mel-spectrograms is clear when we increase the temperature of the two bottom BVAE blocks. On the contrary, the mel-spectrograms are almost the same when we increase the temperature of the top two BVAE blocks. Especially, when we set the temperature of 'BVAE block 2' to 5.0, the mel-spectrograms are the most diverse with good speech quality.

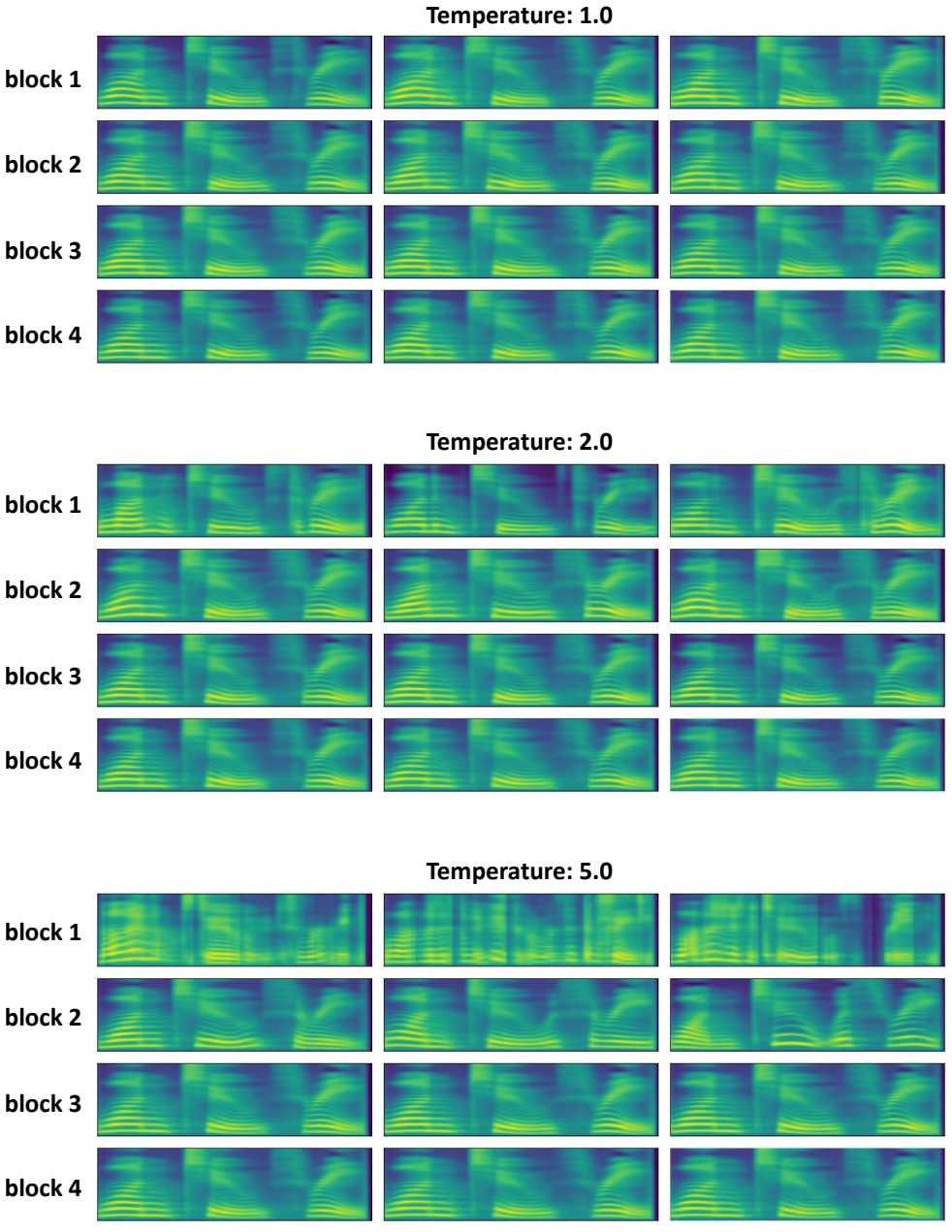

Figure 7: Mel-spectrograms generated from the same text, "One, Two, Three.".

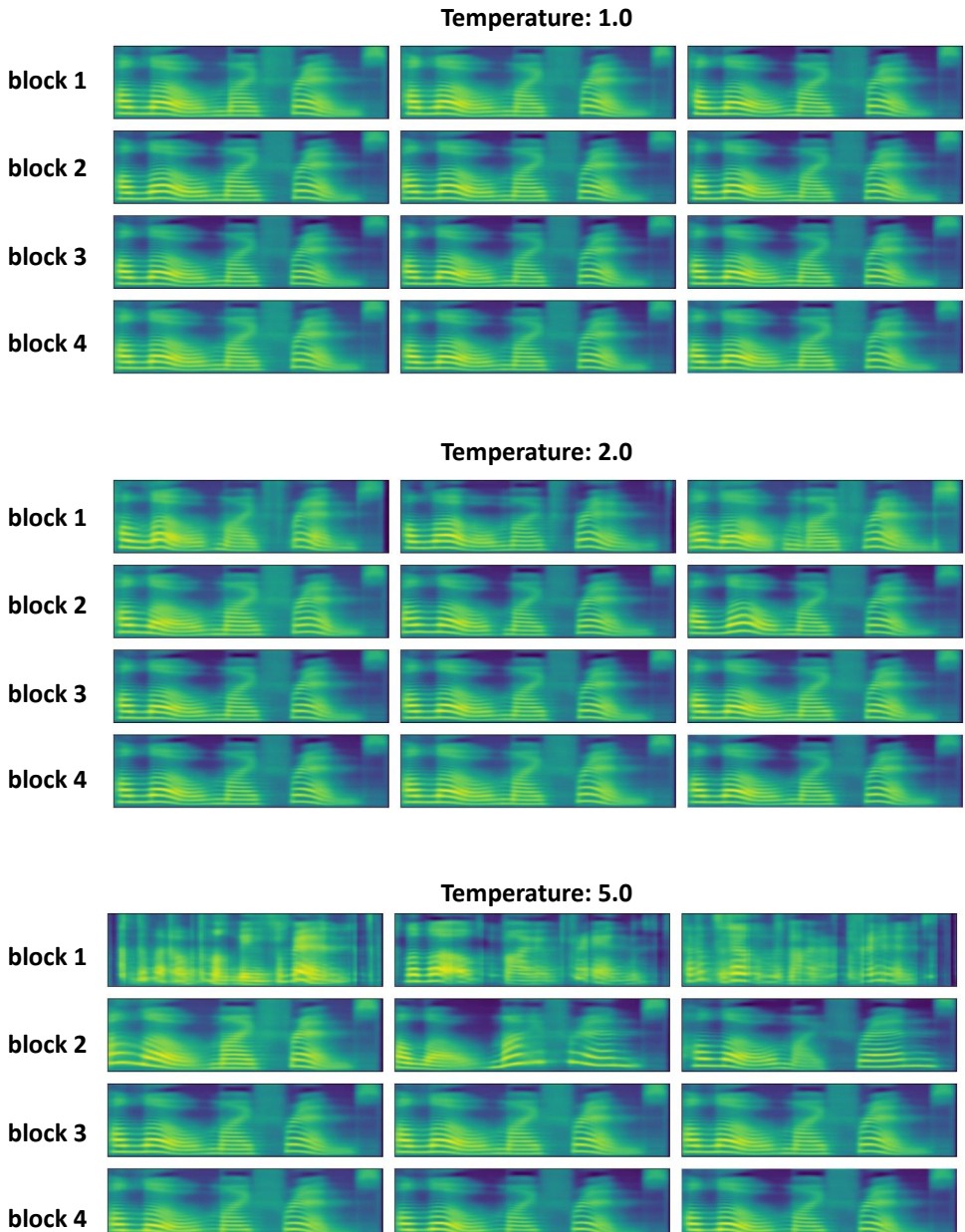

Figure 8: Mel-spectrograms generated from the same text, "Hello, my friends.".

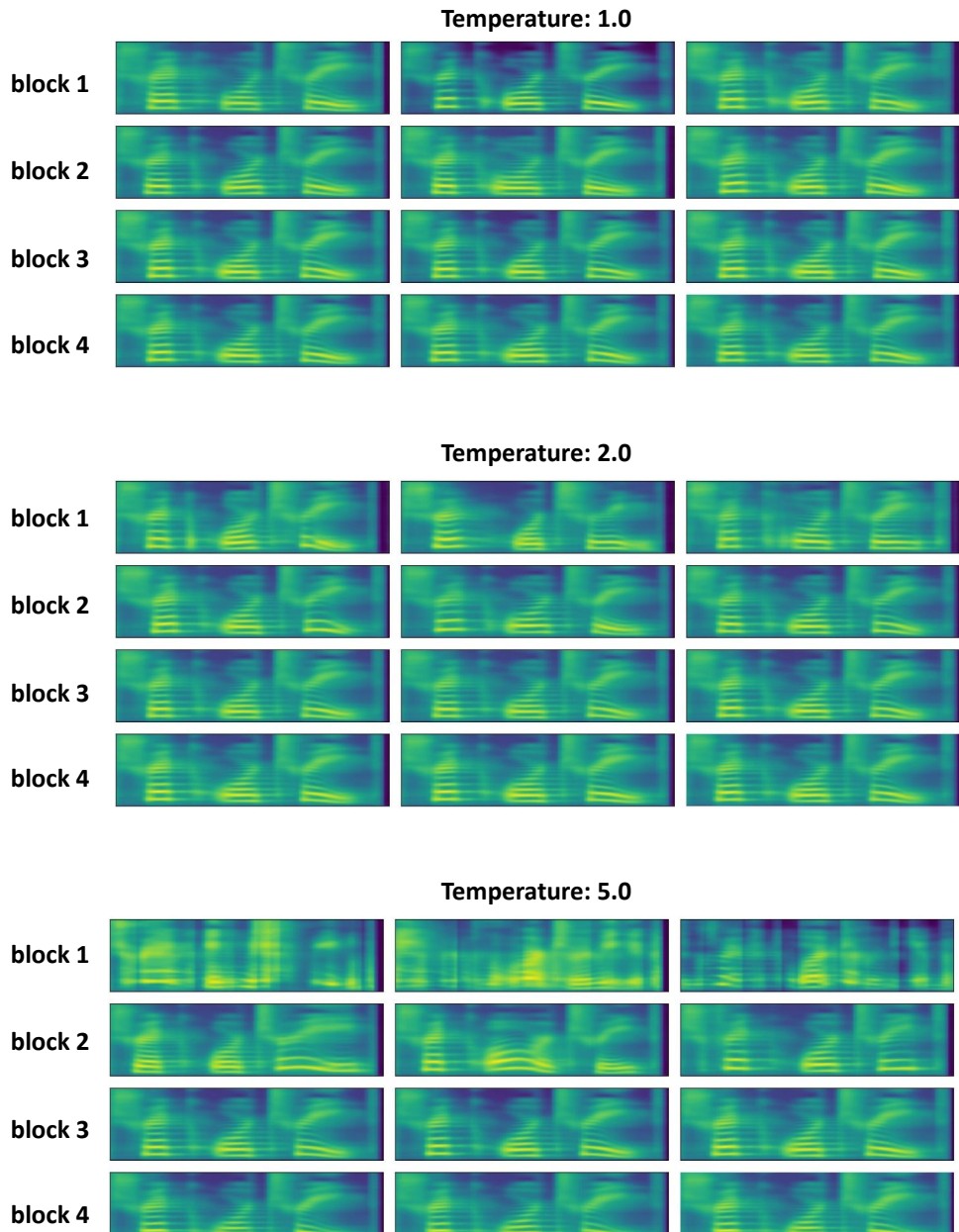

Figure 9: Mel-spectrograms generated from the same text, "Trick or treat!".

## D  SPEED CONTROL

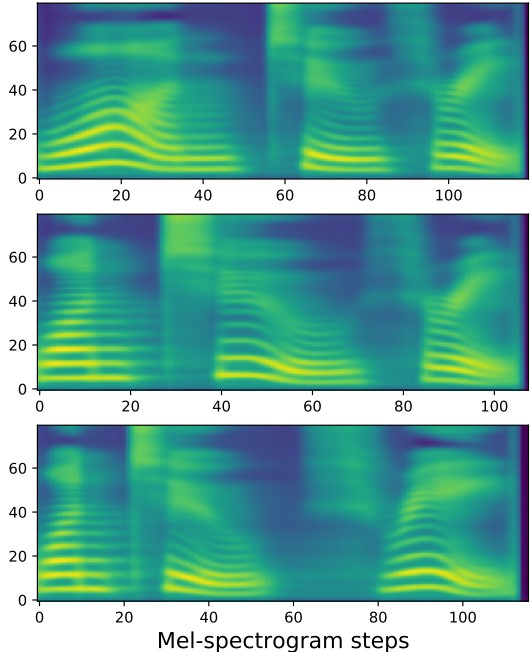

Figure 10: Speed controlled mel-spectrograms generated from the same text "One, Two, Three."

While AR TTS models generally suffer from the lack of controllability, BVAE-TTS can control the fine-grained speed of speech by multiplying a positive constant to the durations predicted by the duration predictor. Figure 10 shows three mel-spectrograms produced by BVAE-TTS using the same sentence, "One, Two, Three.". While changing the target word from "One" to "Three", we multiply 2.0 to the durations of phonemes belonging to the target word, and multiply 0.7 to the durations of phonemes belonging to the non-target words. In this experiment, we observe that BVAE-TTS successfully generates speech while varying the pronouncing speed of each word in a single sentence. Interestingly, we observe that our model intensifies the pronunciation of the target word, showing the capability of appropriately adjusting the prosody according to the speed.

