# OpenReview forum: "Bidirectional Variational Inference for Non-Autoregressive Text-to-Speech"
_ICLR.cc/2021/Conference — ICLR 2021 Poster_

### Official Review · AnonReviewer3 · 2020-10-15
**Great results and thorough evaluation with a well-motivated model, but presentation could be better**

**Rating:** 8
**Confidence:** 4

**Review:**

Summary: Neural models that autoregressively generate mel spectrograms from text (or phonemes), such as Tacotron, have been used to generate high quality synthetic speech. However, they suffer from slow inference speed due to their autoregressive nature. To alleviate this, non-autoregressive models have been proposed, such as FastSpeech and Glow-TTS. The proposed model, BVAE-TTS, is yet another non-autoregressive speech synthesis model (outputting spectrograms), with two key advantages over the aforementioned models: (a) no autoregressive teacher model is required, as in FastSpeech, which simplifies training, and (b) fewer parameters are needed than in Glow-TTS, since there is no bijectivity constraint (allowing a more expressive architecture to be used). Models are compared with inference speed and MOS, and BVAE-TTS compares favorably on both both metrics when compared to Glow-TTS.

Pros:

1. The evaluation of the model is done well, in a clear way. LJSpeech is used, a dataset which is commonly used and easily accessible. MOS and inference speech are provided, and error bars are provided for MOS values. BVAE-TTS is compared to Glow-TTS and Tacotron 2 (one other non-autoregressive model, and one well-known AR baseline), and hyperparameters are provided. A single vocoder (pretrained WaveGlow) is used on all models, isolating the effect of the spectrogram prediction model used.

2. Section 4.3, pertaining to using attention distributions to learn a duration predictor, is interesting and novel. Using positional encodings is standard and using a loss guide is unsurprising. However, while jitter and straight-through estimators are not uncommon, all of these things together make a compelling and novel approach to using attention to infer discretized durations and compensate for that train-test mismatch well. I believe that a similar technique could be used in other models as well.

3. The model is an application of similar ideas from image synthesis, which is interesting, in that it demonstrates that some of those techniques work equally well for spectrogram synthesis. This sort of cross-modal result points to the strength of the method being used, which is a valuable data point for the research community.

Cons:

1. The biggest weakness of this paper, in my view, is that deciphering the model itself is quite difficult. Although the model bears resemblance to NVAE (for which code is released), understanding the fine details is tricky, and the paper does little to aid in that effort.

In particular, understanding the exact layer inputs and outputs and parameters of the normal distributions being used is difficult, and I believe the paper would benefit significantly from a pseudocode explanation of the network. For example, I did not understand why the generative model produced both $\mu_l$ and $\Delta \mu_l$, and whether $\mu_l$ was predicted with a dense layer or was the accumulation of the prior BVAE stacks' $\Delta \mu_l$ values (and similar for $\Sigma$).

I also wonder why the output of the attention layer is not provided to the encoder; perhaps there is a fundamental reason for this which I am missing, or perhaps this is simply an architecture choice.

A very clear explanation of the method itself, perhaps as psuedocode for where the means and variances come from and which features they interact with and what it sampled when, would in my view make this among the top papers.

Recommendation:  Accept. The paper is well written and results are strong, although I would prefer if the method itself were explained more clearly.

---

> ### Author Response · Authors · 2020-11-16
> **Responses to AnonReviewer3**
>
> We thank the reviewer for the great feedback. The feedback is very constructive and helpful for building a better paper.
> Below are our answers to your questions.
>
> Q1. It is quite difficult to understand how the BVAE-TTS works. For example, what are the exact layer inputs and outputs, and how the parameters of the normal distributions are used?
> A1. Thank you for pointing this out, and we also agree that the architecture of BVAE-TTS is quite complicated. In BVAE-TTS, the mean and covariance values are predicted with a 1-D Conv layer (+ softplus). The delta values make the difference between prior and posterior caused by the data observation, and they are not the values accumulated along the layers. Following your suggestion, we will add the pseudo-code explanation of the network in the modified manuscript. Thank you for your suggestion.
>
> Q2.  Why is the output of the attention layer not provided to the encoder?
> A2. If the encoder you mentioned is the bottom-up path of BVAE-TTS, the attention is conducted only at the top of the bottom-up path, instead of between every BVAE block. The output is then inputted directly to the top-down path. We think you have some misunderstandings about this part. However, we think it could be an interesting research to use attention mechanisms between the BVAE blocks, so that the encoder effectively extracts acoustic features that are disentangled to the textual contents. Thank you for your interesting suggestions.

---

### Official Review · AnonReviewer2 · 2020-10-28
**An interesting paper with some nonsolid claims**

**Rating:** 5
**Confidence:** 5

**Review:**

This paper combined fastspeech with a hierarchical VAE (or ladder VAE? in their paper it called bidirectional VAE) to achieve parallel and high quality text-to-mel syntheisis.

The paper claims these contributions: (1) Introducing an online fashion for duration prediction, instead of distillation in FastSpeech and ParaNet. So the model is more e2e. (2) Introducing an BVAE, which extract features hierarchically to better capture prosody (overcome one-to-many) problem. During inference, can use the prior directly. This is directly than previous VAE application in TTS, which is only use to capture residual information. (3) it's faster and with same quality as autoregressive Tacotron and with better quality than other published non-autoregressive model.

The key strength of this paper is the architecture is new. I think using a hierarchical VAE here is reasonable.

My concerns mostly from the conclusion and experiments.
(1) The paper claims compare to previous non-autoregressive model, they are more e2e, since both FastSpeech (also use duration predictor) and ParaNet (without VAE) rely on distillation. However, there is another paper called FastSpeech 2 (https://arxiv.org/abs/2006.04558, published on June 8th), the model also claim " 1) removing the teacher-student distillation to simplify the training pipeline".  Can the author explain the difference? Also i think need to cite that paper because it published in June and very related.
(2) As mentioned in (1), ParaNet and FastSpeech1/2 are very related to this paper. But why only compare with waveglow?
(3) The paper has an ablation study section, but it missing couple very simple baseline. 1) remove VAE, purely predict mel-features based on duration and phoneme embeddings. 2) using a simple VAE instead of hierachical one. How it affect the performance.
(4) One key claim of this paper is that it is as good as Tacotron 2. However, for the in-domain test, the 0.2 behind. By listening the audio samples provided by the author, it indeed significantly worse. The out of domain looks better, I suspect the reason is Tacotron 2 has some attention failures due to it not robust as duration based model. A proper baseline here, is a FastSpeech model. Could you also provide OOD samples? It's really hard to believe such prosody gap can be filled by switch domain.
(5) Back to the original motivation, why we need non-autoregressive model for TTS? For neural based TTS system, most of time is in vocoder. Even we assume the speed for mel-to-spec is important, I don't think measure speed with batch size = 1 is important, because non-autoregressive model can not be streaming. A proper comparison is measure FLOPS and throughput. This might make more sense for offline TTS. This is a minor concern, as long as the quality are good enough.
(6) The paper claims their model is more compact, but there is no comparison for a smaller Taco2 model or other non-autoregressive model.

In summary, based on my understanding, this paper proposed a new non-autoregressive based text-to-mel model with quality regression but possible better robustness. My opinion is that it's a borderline for ICLR, since the importance of the proposed VAE was not well justified, and the quality was not as good as autoregressive model.

---

> ### Author Response · Authors · 2020-11-16
> **Responses to AnonReviewer2 (Part 1/2)**
>
> We thank the reviewer for the extensive comments, which were very constructive and helpful for building a better paper.
> Below are our answers to your questions. Because of the maximum 5000 character limit, we write the answers in two parts.
>
> Q1. What is the difference between your model and FastSpeech 2?
> A1. As you mentioned, FastSpeech 2 said it succeeded in removing the teacher-student distillation. However, to achieve this, it requires additional duration labels and other acoustic features such as pitch and energy information obtained from external tools. On the contrary, since our model only utilizes a text-mel-spectrogram pair, it does not depend on the external tools, and so the training is simpler than the FastSpeech2. We think that the differences will be helpful to clarify the advantages of our model, so we will add this in the modified version with a citation of the FastSpeech2 paper. Thank you for your fruitful question.
>
> Q2. ParaNet and FastSpeech1, 2 are very related to this paper. But why only compare with Glow-TTS?
> A2. We agree that ParaNet and FastSpeech 1,2 can be good baselines for the experiment, but the official source codes for the models are not provided. Therefore, to fairly compare the performance of BVAE-TTS to other TTS models, we choose two models, Tacotron 2 and Glow-TTS, where each represents an AR and a Non-AR TTS model. A pre-trained Glow-TTS model is provided by the author. Although Tacotron 2 from NVIDIA is not provided by the official author, it is widely used and is recognized in the field of speech synthesis as being correctly implemented.
>
> Q3. The paper has an ablation study section, but it is missing a couple very simple baseline: 1) remove VAE, purely predict mel-features based on duration and phoneme embeddings; 2) use a simple VAE instead of hierarchical one.
> A3. When we removed VAE, the TTS model failed to learn mel-spectrogram generation, and when we used simple VAE, the result was the same. Therefore, we worry that the results would be not that informative to be compared with BVAE-TTS. However, if the conclusion of this discussion is that we need to do the further ablation studies, we will report the results in the modified version.
>
> Q4. Tacotron 2 shows better speech quality for in-domain dataset and worse for out-of-domain dataset. However, there are no audio samples generated using out-of-domain texts in supplementary material. Could you also provide out-of-domain audio samples?
> A4. Yes. We will update the supplementary files including the audio samples generated using OOD text data.

---

> ### Author Response · Authors · 2020-11-16
> **Responses to AnonReviewer2 (Part 2/2)**
>
> We thank the reviewer for the extensive comments, which were very constructive and helpful for building a better paper.
> Below are our answers to your questions. Because of the maximum 5000 character limit, we write the answers in two parts.
>
> Q5. Is the non-autoregressive text-to-mel-spectrogram model necessary? For neural based TTS systems, most of time is in vocoder.
> A5. Yes, your point is correct in terms of inference time. However, we think the fact that the inference time does not increase linearly as a text gets longer is still important. Furthermore, non-autoregressive generation shows its strength more in the out-of-domain data, i.e. very long input text, or the text patterns not existing in the training dataset. This is because the AR models suffer from accumulated prediction error. Thank you for your question and we will clarify it.
>
> Q6. Even if we assume the speed for text-to-mel-spectrogram is important, I don't think measuring speed with batch size = 1 is important, because non-autoregressive models can not be used for streaming. A proper comparison is measure FLOPS and throughput.
> A6. We think comparing the inference time of TTS models is more practical to evaluate the models. This is because lower FLOPS for generating a speech does not guarantee shorter inference time. As far as we know, most previous studies on non-autoregressive TTS models also reported their inference time instead of FLOPS or throughput, including ParaNet and FastSpeech 1,2. [1,2,3,4]
>
> Q7. The paper claims their model is more compact, but there is no comparison for a smaller Tacotron2 model or other non-autoregressive model.
> A7. Our initial motivation is to build a new VAE-based non-autoregressive TTS model instead of developing a compact TTS model. Therefore, when we compare the number of parameters, we mainly compare BVAE-TTS and Glow-TTS in terms of both MOS and the number of parameters. This is because both models are non-AR TTS models without a teacher model.
>
> [1]: Ren, Yi, et al. "Fastspeech: Fast, robust and controllable text to speech." Advances in Neural Information Processing Systems. 2019.
> [2]: Kainan Peng, Wei Ping, Zhao Song, and Kexin Zhao. Non-autoregressive neural text-to-speech. In Proceedings of the 37th International Conference on Machine Learning, pp. 10192–10204. PMLR, 2020.
> [3]: Jaehyeon Kim, Sungwon Kim, Jungil Kong, and Sungroh Yoon. Glow-tts: A generative flow for text-to-speech via monotonic alignment search. arXiv preprint arXiv:2005.11129, 2020.
> [4]: Ren, Yi, et al. "FastSpeech 2: Fast and High-Quality End-to-End Text-to-Speech." arXiv preprint arXiv:2006.04558 (2020).

---

### Official Review · AnonReviewer4 · 2020-10-29
**Potentially valuable contribution to parallel TTS, with some concerns.**

**Rating:** 6
**Confidence:** 5

**Review:**

Summary:
This paper presents BVAE-TTS, which applies hierarchical VAEs (using an approach motivated by NVAE and Ladder VAEs) to the problem of parallel TTS.  The main components of the system are a dot product-based attention mechanism that is used during training to produce phoneme duration targets for the parallel duration predictor (that is used during synthesis) and the hierarchical VAE that converts duration-replicated phoneme features into mel spectrogram frames (which are converted to waveform samples using a pre-trained WaveGlow vocoder).  The system is compared to Glow-TTS (a similar parallel system that uses flows instead of VAEs) and Tacotron 2 (a non-parallel autoregressive system) in terms of MOS naturalness, synthesis speed, and parameter efficiency.


Reasons for score:
Overall, I think the system presented in this paper could be a valuable contribution to the field of end-to-end TTS; however, from a machine learning perspective, the contributions are incremental and quite specific to TTS.  In addition, I have some slight concerns about the clarity of the presentation that made it harder to understand the (fairly simple) approach and its motivation than I’d expect from an ICLR paper.  Finally, the quality of the speech produced by the system is only evaluated on a single dataset and uses only 50 synthesized examples in the subjective ratings.  For these reasons, I feel this paper would be a better fit for a speech conference or journal after addressing the evaluation and presentation issues, but I would still support acceptance if other reviewers push for it and my concerns are addressed.


High-level Comments:
* The speed, parameter efficiency, and MOS results are quite promising.  However, when considering the Glow-TTS paper (which this seems like a direct followup to), the system improvements seem quite incremental (replace flows with HVAEs and replace the monotonic alignment search with soft attention plus argmax).
* Incremental system improvements are great if they result in significant improvements that are demonstrated through rigorous experiments, however, compared to Glow-TTS, the experiments are not nearly as comprehensive and convincing. Listening to a few of the audio examples provided in the supplemental materials, I don’t get the sense that the audio quality is significantly better than that of Glow-TTS as is suggested by the MOS numbers (BVAE-TTS sounds a bit muffled to my ears relative to Glow-TTS).
* Since this system uses the same deterministic duration prediction paradigm as Glow-TTS (and other parallel TTS systems), it suffers from the same duration averaging effects and inability to sample from the full distribution of prosodic realizations.
* The motivation would be made clearer if you were more specific early on about the potential advantage of VAE's relative to flows however you want to describe it (parameter efficiency, more flexible layer architectures, more powerful transformations per layer, etc.).
* I'd recommend providing similar motivation for using dot-product soft attention plus straight-through argmax instead of Glow-TTS's alignment search or other competing approaches.  Is it because it's a superior approach or just because it's different from existing approaches?

Detailed Comments:
* Section 2:  I don’t believe Tacotron is actually the *first* end-to-end TTS system.  Maybe it was the first to gain widespread attention, but I know that char2wav (if you count that as e2e TTS) preceded it chronologically in terms of first arxiv submission date.
* Section 2: The Related Work section is fairly redundant with information that is already presented in the introduction.  It might be worth combining the two sections.  This should free up space for additional experiments, explanations, or analysis.
* Section 4.1: The first paragraph here was quite confusing upon a first reading.  I had to read the second sentence (“Via the attention network…”) many times to understand what was being described.
* Section 5.2: I’m curious how you arrived at a sample temperature of 0.333.  Was this empirically tuned for BVAE-TTS or in response to Glow-TTS’s findings?
* Section 5.2, “Inference Time”: It seems important to include details about the hardware platform used to gather the speed results.
* There are minor English style and grammar issues throughout the paper that make the paper slightly more difficult to read.  Please have the paper proofread to improve readability.

Update (Nov 24, 2020):
After reading through the author responses and the updated version of the paper, I feel like a sufficient number of my concerns have been addressed to increase my score to 6.  Specifically, the motivation has been made clearer, the related work section is no longer redundant with the intro, and the authors gave an adequate explanation about the necessity of their attention-based alignment method.

---

> ### Author Response · Authors · 2020-11-16
> **Responses to AnonReviewer4 (Part 1/2)**
>
> Thank you so much for the detailed comments and suggestions. They are really helpful to improve the quality of our paper, especially to make our paper clearer and more convincing.
> Below are the itemized responses regarding each comment. We hope our answers can help our paper sound more convincing to you.
> Because of the maximum 5000 character limit, we write the answers in two parts.
>
> Q1. I have some slight concerns about the clarity of the presentation that makes it harder to understand the approach and its motivation.
> A1. To help the readers understand our motivation and the approach better, we are planning to revise our paper by focusing on making it clearer and making our motivations and advantages more prominent. Furthermore, we will also add a pseudo-code explanation following the R3’s comment in the modified manuscript to improve its understandability.
>
> Q2. The quality of the speech produced by the system is only evaluated on a single dataset and uses only 50 synthesized examples in the subjective ratings.
> A2. We totally understand why you think in that way. Here is our answer. We used the LJSpeech dataset because it is easy to access and many TTS papers also had used only the LJSpeech dataset as a single speaker dataset. In addition, although BVAE-TTS was trained only on the LJSpeech dataset, we evaluated the model on the other 50 out-of-domain sentences to see its generalization ability. When it comes to the number of sentences used for the test, we followed previous TTS papers [1, 2] measuring MOS for about fifty or less sentences.
>
> Q3. When considering the Glow-TTS paper (which this seems like a direct follow-up to), the system improvements seem quite incremental
> A3. We can relate to your concern, however, we think BVAE-TTS is a new direction of non-AR TTS, rather than direct follow-up research of Glow-TTS. In this context, we think our model has so much potential, and we hope that it leads to many improved VAE-based TTS models.
>
> Q4. Listening to a few of the audio examples provided in the supplemental materials, I don’t get the sense that the audio quality is significantly better than that of Glow-TTS as is suggested by the MOS numbers (BVAE-TTS sounds a bit muffled to my ears relative to Glow-TTS).
> A4. Thank you for carefully listening to the audio samples and sharing your impression. When we asked people in my laboratory to listen to the audio samples, many people said they didn’t feel that BVAE-TTS sounds muffled. On the contrary, in terms of naturalness, they said BVAE-TTS is even better than Glow-TTS, e.g. LJ0023-0016, LJ046-0191.
> Also, we think that the muffled sound stands out when the audio samples of BVAE-TTS and Glow-TTS are compared side-by-side. However, since we measured the MOS for the different TTS models independently, we guess BVAE-TTS obtained better MOS than Glow-TTS in terms of naturalness.
>
> Q5. It suffers from the duration averaging effects and inability to sample from the full distribution of prosodic realizations.
> A5. As you point out, we tried to consider the durations also as other latent variables, but it was hard to successfully combine it with an attention mechanism. However, we think it is a very plausible approach and we will study it in future work.
>
> Q6. The motivation would be made clearer if you were more specific early on about the potential advantage of VAE's relative to flows however you want to describe it (parameter efficiency, more flexible layer architectures, more powerful transformations per layer, etc.).
> A6. Thank you for your suggestion. We will revise the introduction and related work sections to clarify the advantages and potential of BVAE-TTS. In the sections, we will add more descriptions about the advantages of BVAE-TTS over the flow-based models that you mentioned.

---

> ### Author Response · Authors · 2020-11-16
> **Responses to AnonReviewer4 (Part 2/2)**
>
> Thank you so much for the detailed comments and suggestions. They are really helpful to improve the quality of our paper, especially to make our paper clearer and more convincing.
> Below are the itemized responses regarding each comment. We hope our answers can help our paper sound more convincing to you.
> Because of the maximum 5000 character limit, we write the answers in two parts.
>
> Q7. I'd recommend providing similar motivation for using dot-product soft attention plus straight-through argmax instead of Glow-TTS's alignment search or other competing approaches. Is it because it's a superior approach or just because it's different from existing approaches?
> A7. Our attention mechanism with ST-argmax is a different approach rather than the improved one of Monotonic Alignment Search (MAS) of Glow-TTS. In terms of the alignment search algorithm, MAS is developed specifically for Glow-TTS. This is because it is trained to maximize the likelihood by directly obtaining the conditional prior distribution of latent representation ‘z’. Since the decoder of BVAE-TTS does not consist of normalizing flows, MAS can not be used in BVAE-TTS. Although it might be possible to use other monotonic alignment search algorithms such as [3], it needs additional dynamic programming computation after the dot-product, and it goes beyond the scope of this study.
>
> Q8. I don’t believe Tacotron is actually the first end-to-end TTS system.
> A8. We missed the paper. We will remove the word ‘first’ and cite the paper too. Thank you for sharing this work.
>
> Q9. The Related Work section is fairly redundant with information that is already presented in the introduction.
> A9. We are thinking of clarifying the advantages of BVAE-TTS over the flow-based TTS models and other previous TTS models in the introduction and related work sections. We will consider your suggestion and reflect it in the revised version.
>
> Q10. The first paragraph of Sec 4.1 is quite confusing upon a first reading. I had to read the second sentence (“Via the attention network…”) many times to understand what was being described.
> A10. Thank you for pointing this out. We will edit the part to help the readers understand more clearly, especially focusing on the second sentence.
>
> Q11. I’m curious how you arrived at a sample temperature of 0.333. Was this empirically tuned for BVAE-TTS or in response to Glow-TTS’s findings?
> A11. We chose the temperature 0.333 after listening to the samples generated with different temperatures, 0, 0.333, 0.6, 1.0. As the qualities are not that sensitive to the temperatures, we unified the temperature to 0.333 following the Glow-TTS. (It showed the best performance on LJSpeech in Glow-TTS.)
>
> Q12.“Inference Time”: It seems important to include details about the hardware platform used to gather the speed results.
> A12. We described our hardware setting in Section 5.1, but not mentioned how we use the hardware setting to measure the inference time, i.g. Are the inference times measured on CPU or GPU?. We will add the description in the revised version.
>
> Q13. There are minor English style and grammar issues throughout the paper that make the paper slightly more difficult to read. Please have the paper proofread to improve readability.
> A13. We will make much more effort to improve the readability of our paper by having much more extensive proofreading.
>
> [1]: Kim, Jaehyeon, et al. "Glow-TTS: A Generative Flow for Text-to-Speech via Monotonic Alignment Search." arXiv preprint arXiv:2005.11129 (2020).
> [2]: Miao, Chenfeng, et al. "Flow-TTS: A Non-Autoregressive Network for Text to Speech Based on Flow." ICASSP 2020-2020 IEEE International Conference on Acoustics, Speech and Signal Processing (ICASSP). IEEE, 2020.
> [3]: He, M., Deng, Y., He, L. (2019) Robust Sequence-to-Sequence Acoustic Modeling with Stepwise Monotonic Attention for Neural TTS. Proc. Interspeech 2019, 1293-1297, DOI: 10.21437/Interspeech.2019-1972.

---

### Official Review · AnonReviewer1 · 2020-10-30
**Novel, fast architecture with many insightful ideas - accept**

**Rating:** 6
**Confidence:** 5

**Review:**

Post rebuttal and discussion
========================
Several reviewers have pointed out that the paper needs more comparisons/ablations with existing models (e.g. Paranet/Fastnet). To this end, I think we at least need a comparison with Paranet, which is a 'comparable' non-autoregressive CNN based VAE based model with a few other components such as attention distillation.

There are components in the paper that could do with more ablation studies
- argmax with straight through estimator
- some guidelines on BVAE blocks and tuning

In light of these points, together with the fact that we don't have any theoretical novelties in this paper, I reduce my score to 6. Even so, I feel that the paper would be a valuable contribution because
a) A generative model (GAN/VAE/VQVAE/Flow based models/score matching based models) might add extra benefit in the synthesis problem, as compared with a supervised model without a similar generative component such as Tacotron. The NVAE has been shown to significantly outperform the regular VAE in image generation tasks. It stands to reason that it would do well in speech generation also.
b) Speed, robustness and ease of implementation (although this remains to be demonstrated).

Initial Review
===========

This paper proposes a non-autoregressive (non AR) way to perform text to speech synthesis. It uses a VAE based setup - adapted from the recent image paper NVAE to build two stacks of hierarchical VAE blocks  (in priors), one going bottom up and the other, top down. The key claims are that it results in improved speed, and reduced model footprint from using a non AR architecture, with excellent quality comparable to the best autoregressive/recurrent methods in Tacotron2 [2] and non AR glow-TTS[3].

The work contains many interesting ideas for TTS, and I am very interested in seeing how this work pans out in practical speech synthesis applications.

Key ideas:
1. The bidirectional stack, which they call BVAE is adapted from the recent NVAE work which has produced stellar image generations. The model uses 1D convolutions under the hood, in contrast with the fashionable, but slow autoregressive flows or recurrent models. If one can get such a model to work, it could be advantageous in effecting savings in computational time and model size.

During training, at the top of the bottom-up stack, text features are inflated to the size of the mel spectrogram features, and reconstructed with the top down BVAE stack. For inference, text is inflated to an expanded text matching audio mels, and then sent down the top-down stack to give a mel sample.

2. Attention modeling: An important consideration here is to align text and mel, commonly done with an attention mechanism. In this work, the attention alignment shows up as a duration model, which is rather interesting, and seemingly gives additional flexibility. After aligning text and mel (using dot product), the alignment can be reinterpreted as a duration model by comparing phoneme and mel frame alignments. Furthermore, they use a discrete match with argmax rather than a sum over all attention alignments as is generally done. This also necessitates the use of the straight-through estimator while backpropagating since the durations are rounded entities. This type of modeling seems also to be used in the Glow-TTS  work but with alignments determined through dynamic programming.

I found the result that the model is not very sensitive to alignment mismatches to be quite remarkable.

3. Fittings for robustness during inference: They use several instructive ideas - jittering text, adding positional embeddings, diagonal penalty (since alignment is mostly diagonal) and KLD annealing.

4. Analyses - ablations to see which of the VAE blocks affect the result by varying temperature (from Glow [3]).

My thoughts:
Generally, the paper made for fascinating reading. Having worked with Tacotron, I have always felt that adding a VAE to that (RNN based) setup would improve its generative capabilities by giving it additional regularization qualities, among other things. That we can see the model perform better when we add jitter and can also respond to the duration specified seems to corroborate that in a loose way (figure 10).

- Could the authors clarify how the duration modeling results in 'monotonic' alignments? As far as I can see, the argmax guarantees a unique match, but is monotonicity necessary?

From section 5.3.2:
"Since the text is forced to be used monotonically in the duration-based generation, it makes the model more robust to the attentionerrors while making fewer pronouncing mistakes."

- A comparison with an equivalent soft attention implementation might be insightful.

- Multi Speaker TTS: I am wondering how this model would perform in a multispeaker dataset, say libritts. One aspect that the paper does not touch in detail is in its capabilities as a generative model. It would be interesting, for instance, to see if this model can in any way separate speaker style from content with a multispeaker model.

Overall, I think this paper would be a good addition to the body of speech synthesis work, and recommend that it is accepted.


[1] NVAE: https://arxiv.org/pdf/2007.03898.pdf
[2]: Tacotron2: https://arxiv.org/pdf/1712.05884.pdf
[3] Glow-TTS: https://arxiv.org/pdf/2005.11129.pdf
[4]: Glow: https://arxiv.org/pdf/1807.03039.pdf

---

> ### Author Response · Authors · 2020-11-16
> **Responses to AnonReviewer1**
>
> We thank you for your interest in our research and we also hope it becomes a good starting point for VAE-based TTS research.
> Below are our answers to your questions.
>
>  Q1. How does the duration modeling result in 'monotonic’ alignment?
>  A1. ‘Monotonic alignment’ means phoneme representations are used in an orderly manner. In other words, there is no case where a phoneme representation that appears later in a sentence is used earlier in the decoder. In this context, if we inflate the phoneme representations based on their durations, the above situation never happens. Thank you for your fruitful question and we will clarify this in the revised version.
>
> Q2. A comparison with an equivalent soft attention implementation might be insightful.
> A2. It is the situation where we train BVAE-TTS without using ST-argmax technique. (Sec 5.3.2.) When we remove the constraint of a one-to-one mapping between phonemes and mel-spectrogram frames, our model fails to learn the alignment. Thank you for pointing this out and we will clarify this in the revised version.
>
> Q3. I am wondering how this model would perform in a multi-speaker dataset. One aspect that the paper does not touch in detail is in its capabilities as a generative model. It would be interesting, for instance, to see if this model can in any way separate speaker style from content with a multi-speaker model.
> A3. Thank you for your suggestion, however, our initial aim was to develop a novel TTS model based on VAE architecture, and so we focus on succeeding in generating speech with competitive quality. However, we are actually planning to extend our model to the multi-speaker scenario in future work. For example, we expect that, by letting the model extract a global latent vector from a mel-spectrogram, the model can change the global style of the speech (e.g. speaker identity) by controlling the global latent vector.

---

### Author Response · Authors · 2020-11-18
**The revised paper is uploaded. (Last edited Nov 23 21:05 AoE)**

We have uploaded a revised paper to incorporate the reviewers' comments, concerns, and suggestions.
Thank all the reviewers for their constructive comments and extensive analysis that are really helpful to make our paper more complete.

Specifically, the updated version includes:
* We have modified the configuration of the paper focusing on clarifying the motivation and advantages of BVAE-TTS.
* We have done much more extensive proofreading to improve its readability and we have tried to help readers understand our approach by adding more explanations, including the pseudo-codes for training and inference of BVAE-TTS in the appendix section.
* Supplementary material has been updated including the audio samples for MOS-OOD.
* Minor typos and inconsistent reference format have been fixed in the revised version. (Last edited Nov 23 21:05 AoE)

---

### Decision · Program_Chairs · 2021-01-07
**Final Decision**

**Decision:**

Accept (Poster)

**Comment:**

Non autoregressive modelling for text to speech (TTS) is an important and challenging problem. This paper proposes a deep VAE approach and show promising results. Both the reviewers and the authors have engaged in a constructive discussion on the merits and claims of the paper. This paper will not be the final VAE contribution to TTS but represents a significant enough contribution to the field to warrant publication. It is highly recommended that the authors take into account the reviewers' comments.